# Time-to-recovery from severe pneumonia and its predictors among children 2–59 months of age admitted to the pediatric ward of Jimma University Medical Center, Southwest Ethiopia, 2023: A retrospective cohort study

**Getu Girma Bekele**[1]*, **Tola Getachew Bekele**[1], **Masrie Getnet**[2], **Dawit Regassa**[2]

1 Jimma University Medical Center, Jimma, Ethiopia, 2 Jimma University, Institute of Health, Faculty of Public Health, Department of Epidemiology, Jimma, Ethiopia

* getgirma2013@gmail.com

## Abstract

### Background

Pneumonia is an inflammation of lung parenchyma. The World Health Organization estimated 156 million cases of pneumonia occur annually. Out of them, 20 million cases severe enough to require hospitalization, and each year 1.2 million deaths occur among under-five children. Despite studies and initiatives aimed at reducing pneumonia related deaths in children, Ethiopia is ranked sixth among top fifteen countries in terms of pneumonia related morbidity and mortality.

### Objectives

This study aimed to assess the time to recovery from severe pneumonia and its predictors among children aged 2–59 months admitted to the pediatric ward of Jimma University Medical Center; Southwest, Ethiopia, 2023.

### Methods

A facility-based retrospective cohort study was carried out among 426 children aged between 2 and 59 months. Five years of medical records, from 2018–2022, were reviewed. A simple random sampling technique was used. Data entry was done in Epidata version 4.6 and exported to and analyzed by STATA version 15. Variables with p-value < 0.25 at Bivariable Cox regression analysis were selected for the multivariable Cox proportional model. A multivariable Cox regression model with 95% confidence interval and Adjusted Hazard Ratio was used to identify a significant predictor of time to recovery at a p-value < 0.05.

**Data availability statement:** All relevant data are within the manuscript.

**Funding:** The author(s) received no specific funding for this work.

**Competing interests:** The authors have declared that no competing interests exist.

**Abbreviations:** AHR, Adjusted Hazard Ratio; BSc, Bachelor of Science; CEO, Chief Executive Officer; CI, Confidence Interval; CHR, Crude Hazard Ratio; FMOHE, Federal Ministry of Health of Ethiopia; HIV, Human Immune Virus; Hib, Haemophilus influenzae type b; ICCM, Integrated Community Case Management; IMNCI, Integrated Management of Newborn and Childhood Illness; IQR, Interquartile range; JUMC, Jimma University Medical Center; MRN, Medical Registration Number; ORS, Oral Rehydration Solution; PCV, Pneumococcal Conjugate Vaccine; SAM, Severe Acute Malnutrition; SD, Standard Deviation; SSA, Sub-Saharan Africa; TB, Tuberculosis; WHO, World Health Organization.

## Result

The median recovery time was 4 days (IQR: 3, 7). Incidence rate of recovery was 15.78 per 100-person day (95% CI 14.2–17.5). The presence of co-morbidity (AHR; 0.7, 95% CI (0.54–0.91)), being treated with Ceftazidime and Vancomycin (AHR; 0.29, 95% CI (0.14–0.60)), antibiotic change (AHR; 0.74, 95% CI (0.58–0.95)) and late presentation to the Hospital (AHR; 0.58, 95% CI (0.43–0.78)) were statistically significant predictors that prolong recovery time.

## Conclusion

The median recovery time was longer than other similar studies. Therefore, due attention should be given to the identified predictors of the recovery time.

## Introduction

The World Health Organization estimated that 156 million cases of pneumonia occur annually in children under age of five, including up to 20 million cases severe enough to require hospitalization, and 1.2 million deaths each year [1]. The greatest incidence was occurring in south Asia (2,500cases per 100,000 children) and west and central Africa (1,620 cases per 100,000 children) [2]. Approximately 490,000 children under the age of five died of pneumonia in Sub Saharan Africa (SSA) in the year 2016 [3]. In Ethiopia, approximately 3,370,000 children contract pneumonia each year which accounts for 14% of all deaths of children under five years old [4].

Severe pneumonia is the major cause of hospitalization in under-five children. Among 156,847 admitted children identified in seven hospitals in Bangladesh, pneumonia was the most diagnosed (17.9%) [5]. Hospitalization alone was found to contribute to 46.8% of a household price for one severe pneumonia episode [6]. Effective management of pneumonia is not just only the ratio of the children recovered, but also the time of recovery. Spending each night in the hospital increases the risk of developing adverse medication reactions, ulcers, and infections by 0.5%, 0.5%, and 1.6%, respectively [7]. Tremendous independent clinical factors marked up mortality and duration of stay associated with pneumonia, with significant variability among hospitals. Some studies identify different forms of malnutrition, co-morbidities, diarrhea, antibiotics first administered, being underweight, and age are the major factors that affect recovery time [8].

A longer hospital stay has a significant negative impact on the admitted child, his or her parents, and the healthcare system, particularly in societies with resource-limited settings and no health insurance scheme [9]. Between 26.6% and 115.8% of monthly household income is spent on the direct medical costs of treating severe pneumonia in a hospital setting. Non-medical costs (such as travel and food) during a child's hospital stay for severe pneumonia account for 9.0% and 31.0% of the household's monthly income in Kenya and Guinea, respectively [10]. Patients with prolonged length of stay extremely account for the consumption of more hospital resources, Hospital acquired infection, and denies critically ill patients' timely access to treatment and contributes to capacity shortages [11]. Hence, these children are often discharged against medical advice without receiving full therapy, which results in an increased rate of readmission and complications such as pneumonic effusion, empyema, necrotizing pneumonia, and lung abscess [12].

The Federal Ministry of Health of Ethiopia (FMOHE) has been working to prevent and reduce under-five mortality by introducing Pneumococcal Conjugate Vaccine (PCV)

and Haemophilus influenzae type b (Hib) vaccines. The ministry is also implementing key strategies such as integrated community case management (ICCM) to train, support, and supply community health workers to enhance diagnostic, treatment and referral services and integrated management of newborn and child illnesses (IMNCI) to improve case management skills of health care providers, improving health systems to provide quality care and improving family and community health practice for health, growth, and development [13,14]. Ethiopian Health Sector Transformation Plan in 2015, to reduce the mortality rate of children to 31 and 41 per 1000 live births in 2025 and 2035 respectively [15]. Despite these program intervention, pneumonia remain the major cause and of illness and death in children under age of five in Ethiopia; with an approximated contribution of around 40,000 fatalities per year [16].

Pneumonia is one of the major public health importance. Despite the intervention that has been implemented in Ethiopia, pneumonia continues as one of the main reasons of hospitalization and mortality among under-five children [9]. The widespread nature of the problem in Ethiopia has already killed thousands of children hospitalized with severe pneumonia, which requires everlasting solutions [17]. Understanding the average length of stay in hospitals helps the health managers in evaluating the quality of health care delivered by a given health institution because the average length of stay indicates information about efficiency of care service delivery, and the standard of care being provided, and effective use of resources. Furthermore, it can also provide valuable information to physicians and other healthcare providers to counsel parents or caregivers about the duration of hospitalization so that the caregivers can timely mobilize resources for a child's hospital care.

To the best of our knowledge and as far as our literature search is concerned, there are limited studies and evidence on recovery time from severe pneumonia and its predictors in the current study area. As a result, the aim of this study was to estimate the median time of recovery from severe pneumonia and its predictors in children aged 2–59 months admitted to the pediatric ward of Jimma University Medical Center, Southwest, Ethiopia, 2023.

## Materials and methods

### Study area and period

The study was conducted at Jimma University Medical Center (JUMC). It is the only teaching and tertiary federal hospital found in southwest Ethiopia. Geographically, it is located in Jimma town, Oromia region, 353 km southwest of Addis Ababa, the capital city of Ethiopia. It was established in 1930 E.C. by Italian invaders for the service of their soldiers. Currently, JUMC provides services for approximately 18,289 inpatients on 800 beds, 232,000 outpatient attendants, 79,000 emergency service cases, and 6,500 deliveries in a year coming to the hospital from the catchment area. JUMC serves as a referral hospital in southwest Ethiopia. The population in the catchment area is estimated to be 15 million [18,19].

The pediatric department of JUMC has a pediatric outpatient, pediatric ward, and pediatric intensive care unit. The pediatric ward has three units namely level I, level II, and nutritional rehabilitation unit (NRU) with a total of 78 beds, and provides services for an average of 55–65 patients including severe pneumonia. On average over 400 children were diagnosed with severe pneumonia each year in this hospital. Children diagnosed with severe pneumonia were directly admitted to level I and treated by senior Pediatricians, Residents, Nurses, and Medical Interns. The pediatric ward has a total of 60 health professionals with 35 nurses, 7 senior physicians, and 16 residents. There are also pharmacy professionals and laboratory technologists who were working there by rotation [19]. The five-year data were reviewed from January 1, 2018, to December 31, 2022. The patient charts were retrieved from May 14- June 15, 2023.

## Study design

A facility-based retrospective cohort study was conducted.

## Population

**Source population.** All children 2–59 months of age who were admitted to the pediatric ward of Jimma University Medical Center (JUMC) with severe pneumonia were the target population.

**Study population.** All children with severe pneumonia aged 2–59 months who were managed at the pediatric ward of Jimma University Medical Center (JUMC) from January 1, 2018- December 31, 2022 were the study population.

## Eligibility criteria

**Inclusion criteria.** All children with severe pneumonia aged 2–59 months who were admitted to the pediatric ward of Jimma University Medical Center (JUMC) from January 1, 2018- December 31, 2022, were included.

**Exclusion criteria.** Children's charts with no Medical Registration Number (MRN), admission date, discharge date, and no or ambiguous age were excluded.

## Sample size determination

The sample size was calculated by using the Schoenfeld formula for survival analysis by taking the following assumptions into account [20].

Significance level (α) (two-sided) =0.05, $Z_{\alpha/2} = 1.96$

- Power = 80%, $\beta = 0.2$  $Z_{\beta=}$ 0.842

- AHR = Adjusted hazard ratio was used from different similar studies

- E = number of events (recovery)

- Pr(E) = probability of recovery was used from different similar studies

- Finally, a 10% non-response rate (NRR) for an incomplete chart was added

- $E = (z\alpha + z\beta)2/(\ln \text{AHR})^2 P (1\text{-}P)$, n = E/Pr(E) [20]

- The final largest sample size **426** was considered to conduct the study.

## Sampling technique and procedure

**Sampling technique.** A simple random sampling technique was used to select the study populations.

**Sampling procedure.** The registry logbook of all children who were admitted with severe pneumonia to the pediatric ward of Jimma University Medical Center (JUMC) from January 1, 2018 – December 31, 2022, was reviewed. Since there was no case-specific registration log book, the list of all children aged 2–59 months admitted with severe pneumonia from January 1, 2018 – December 31, 2022, was prepared by excluding those with no MRN, admission date, discharge date, incomplete and ambiguous age. A total of 1985 children with severe pneumonia were noted from the registration logbook during the study period, of which 1697 completed charts were available. Then, 426 charts were sampled using a simple random sampling technique via a computer-generated method (Fig 1).

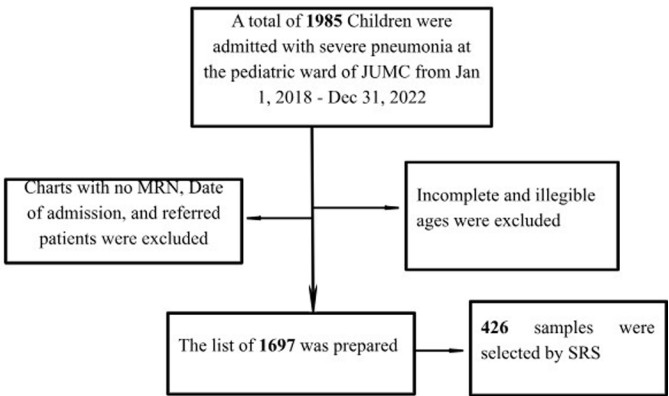

**Fig1. A schematic representation of the sampling procedure to select the study population.**

## Data collection tool and procedure

**Data collection tool.** The data were collected by using a structured checklist adapted from previous similar studies [8,12,21,22]. The tool was designed based on the study objectives and contains socio-demographic characteristics, clinical and treatment-related factors, vaccination and nutritional status as well as co-morbidities.

**Data collection procedure.** The unique medical record number of each patient was used to get their records (charts). All available information on the patient chart was checked. Then all the relevant variables were extracted from patient charts using an English version data extraction tool to meet the study objectives. Two skilled BSc nurses working in the pediatric ward of Jimma University Medical Center (JUMC) extracted the data from patient charts under the supervision of an MSc holder in pediatric and child health working in the pediatric ward of JUMC as a Head Nurse. Training was given to data extractors and supervisor regarding the significance of the study and ways of the data extraction process by the principal investigator.

## Study variables

**Outcome variable.** Time to recovery from severe pneumonia measured in days.

**Independent variables.** *Socio-demographic characteristics*: The socio-demographic characteristics include the age, sex of the children, residence, and health insurance scheme. *Clinical and treatment-related factor:* Under clinical and treatment-related factors Presence of Danger signs at admission, the duration time before seeking health care, Grunting, Head nodding, HIV status, Antibiotics first administered, Antibiotic change, and Convulsion were used.

*Nutrition and Vaccination status: Nutrition and Vaccination status include Vaccination status, height for age, Weight for height, weight for age, Exclusive breastfeeding, Inability to feed, and different co-morbidities.*

## Operational definition

**Severe pneumonia.** children presented with inability to drink, persistent vomiting, oxygen saturation less than 90%, convulsion, and lethargy or reduced level of consciousness [23].

**Time to recovery.** The time from admission to when the child is discharged from the hospital due to improvement in symptoms calculated by number of days.

**Median recovery time.** The time in which half (50%) of the children had recovered.

**Censored.** A child with severe pneumonia who has been admitted to the ward and happens to be dead, self-discharged, or with unknown outcome status.

**Comorbidity.** Any disease acute or chronic present during admission in addition to severe pneumonia.

**Danger signs.** Children with difficult breathing, chest in drawing, oxygen saturation less than 90%, unconsciousness [24].

**Vaccination status.** Fully Vaccinated which refers to Children who had received all forms of vaccination; Partially Vaccinated (Children who had taken at least one PCV dose) and Unvaccinated (Children who have never received PCV or any form of vaccination).

**Discharge status.** indicate the condition (reason) for the patient's termination from the program.

**Exclusive breastfeeding.** Children who were introduced to only breast milk except for drugs, ORS (Oral Rehydration Solution), during the first six months of age [25].

**Underweight.** Children having weight for age is less than -2SD (standard deviation) [26].

**Wasting.** Children having weight for height is less than -2SD (standard deviation) [27].

**Stunting.** Children having height for age is less than -2SD (standard deviation) [27].

## Data management

**Data quality assurance.** Data collectors were given orientation on the content of the data extraction tool and procedure. Two skillful BSc nurses working in the pediatric ward of Jimma University Medical Center (JUMC) reviewed the records after they were given training by the principal investigator. Prior to actual data collection, the data extraction checklist was tested on 21 medical charts at Shenen Gibe Hospital to ensure that the data extraction format is agreed with the objectives of the study. During the data collection process, regular supervision was performed by the supervisor to ensure the quality of data. Throughout the data collection process, the primary investigator and the supervisor routinely examined the accuracy and completeness of the data collected. Data entry and cleaning were done daily and timely feedback was being provided to the data collectors.

**Data analysis procedure.** The data were entered in Epidata version 4.6 exported to and analyzed using STATA version 15. The data were coded and cleaned prior to analysis. The data were described in terms of central tendency (median) and dispersion (interquartile range) for continuous data and frequency distribution for categorical data and presented using graphs, tables, and text. A day was used to calculate the recovery time. Kaplan Meier's plot was used to compare the survival experience of different groups of patients by using survival curves. The association between the independent variable and time to recovery from severe pneumonia was analyzed using the Cox proportional hazard regression. Log rank test was used to identify the presence of significant difference in median survival time between different groups [28].

Univariate analysis was performed and predictors that have an association with the outcome variable at a p-value of 0.25 or less were entered into the multivariable Cox regression model. The presence of multicollinearity was checked using the variance inflation factor (VIF). A Variance Inflation Factor value of < 10 was used as a cutoff point for indicating the absence of multicollinearity. A Cox proportional hazard assumption was checked statistically using the global residual test. The final model adequacy was tested by using the Cox-Snell residual test and the baseline distribution of the cumulative hazard function of residuals against Cox-Snell residuals was follows 45 degree indicating adequately fitted data. Adjusted hazard ratio (AHR) with a 95% confidence interval (CI) was estimated by multivariable Cox

regression and a p-value less than or equal to 0.05 was used to declare the presence of statistically significant association between recovery time and explanatory variable.

## Results

### Socio-demographic and treatment-related characteristics of study population

The study included records of children 2–59 months of age admitted to the pediatric ward of Jimma University Medical Center (JUMC) with a diagnosis of severe pneumonia for five consecutive years (2018–2022). From the sample of 426, information was extracted from 394 patient charts and gives a response rate of 92.5%. The female-to-male ratio was 1:1.21 and 52.28% of the study participants came from the countryside. The median age of the study participants was 10 months (IQR: 5–21) and the majority (218: 55.33%) of them were under the age group of 2–11 months. Almost two third (244: 62%) of the study participants were not enrolled in health insurance schemes.

About 47.72% of the children visited the hospital within three days of onset of illness. Danger signs were present in about (135: 34.26%) of the children at admission. The commonest danger signs were difficult breathing and severe chest in-drawing accounting for (30: 7.61%) and (25: 6.35%) respectively. Among 394 children (188: 47.72%) and (71: 18.02%) had grunting and convulsion respectively. Crystalline penicillin was given to (166: 42.35%) of the children at admission as a first-line treatment and antibiotics were changed for (128: 33%) of the children (Table 1).

### Vaccination, nutritional status, and co-morbidities of the study populations

Majority (281: 71.32%) of the children had a history of exclusive breastfeeding. Of the study participants, about (166: 42.13%) of the children were fully vaccinated whereas (60: 15.23%)

**Table 1. Frequency distribution of socio-demographic and treatment-related characteristics of children aged between 2-59 months with severe pneumonia admitted to the pediatric ward of Jimma University Medical Center (JUMC) from Jan 1, 2018-Dec 31, 2022(n = 394).**

| Variable | Category | Status | |
|---|---|---|---|
| | | Recovered (%) | Censored (%) |
| Age in month | 2–11<br>12–35<br>36–59 | 196 (49.7)<br>111 (28.2)<br>42 (10.7) | 22 (5.6)<br>17 (4.3)<br>6 (1.5) |
| Sex | Male<br>Female | 191 (48.5)<br>158 (40.1) | 25 (6.3)<br>20 (5.1) |
| Danger signs | No<br>Yes | 238 (60.4)<br>111 (28.2) | 21 (5.3)<br>24 (6.1) |
| Residence | Urban<br>Rural | 170(43.16)<br>179(45.43) | 18(4.56)<br>27(6.85) |
| Antibiotics change | No<br>Yes | 229 (59.0)<br>114 (29.4) | 31 (7.99)<br>14 (3.61) |
| Antibiotic first given | Crystalline penicillin Ampicillin and gentamicin Ceftriaxone Ceftazidime & vancomycin | 152 (38.78)<br>77 (19.64)<br>109 (27.81)<br>9 (2.3) | 14 (3.57)<br>12 (3.06)<br>16 (4.08)<br>3 (0.77) |
| Grunting | No<br>Yes | 179 (45.4)<br>170 (43.19) | 27 (6.85)<br>18 (4.56) |

were unvaccinated. Stunting, wasting, and being underweight were found in (143: 36.29%), (186: 47.21%), and (161: 40.86%) of the study participants respectively. About (82: 20.81%) of the children were born before the expected date of delivery. Almost (201:51%) of the children tested non-reactive for HIV and (6: 1.53%) of the children among the study participants were found reactive for the HIV test. About (98: 24.87%) of the children admitted with severe pneumonia had different co-morbidities. Severe acute malnutrition was one of the most common diseases in addition to severe pneumonia among the study participants. Other medical conditions like Bronchial asthma, Down syndrome, and pertussis accounted for 9 (2.29%) (Table 2).

## Treatment outcome and incidence rate of recovery

Regarding the treatment outcome of severe pneumonia, the majority (349: 88.58%) of children recovered, (22: 5.58%) died, (14: 3.55%) were left against medical advice and (9: 2.28%) were discharged with unknown condition (Fig 2).

The shortest and longest length of hospital stay was 1 and 29 days respectively. The total person-day (person time risk) contributed by the study participants was 2,211 days. The overall incidence rate recovery from severe pneumonia was 15.78 per 100-person day (95% CI, 14.2–17.5) observation.

## The median time to recovery from severe pneumonia

The Kaplan-Meier survival curve was used to estimate the survival status of the children with severe pneumonia. According to the Kaplan-Meier survival estimation, the median survival time without using any stratification factors was 4 days (IQR: 3, 7). The graph shows a tendency to fall off quickly in the first five days, showing that most children recovered from severe pneumonia during this time period (Fig 3).

Kaplan Meier survival plot was performed to compare survival probability between categories of different predictors. Accordingly, survival probability among different categorical predictors such as co-morbidities, duration of time before seeking health care, weight for age, antibiotics first administered, and antibiotic change were different (Fig 4).

**Table 2. Frequency distribution of vaccination, nutritional status, and co-morbidities of children aged between 2-59 months with severe pneumonia admitted to the pediatric ward of Jimma University Medical Center (JUMC) from Jan 1, 2018-Dec 31, 2022(n = 394).**

| Variable | Category | Status | |
|---|---|---|---|
| | | Recovered (%) | Censored (%) |
| Vaccination status | Fully vaccinated | 152 (38.58) | 14 (3.55) |
| | Partially vaccinated | 151 (38.32) | 17 (4.33) |
| | Unvaccinated | 46 (11.67) | 14 (3.55) |
| Weight for height | Normal | 183 (46.45) | 25 (6.34) |
| | Wasted | 166 (42.13) | 20 (5.08) |
| Exclusive breastfeeding | No | 99 (25.13) | 14 (3.55) |
| | Yes | 250 (63.45) | 31 (7.87) |
| Weight for age | Normal | 210 (53.29) | 23 (5.84) |
| | Underweight | 139 (35.27) | 22 (5.58) |
| Height for age | Normal | 219 (55.58) | 32 (8.14) |
| | Stunted | 130 (32.99) | 13 (3.29) |
| Comorbidities | No | 262 (66.5) | 34 (8.6) |
| | Yes | 87 (22.1) | 11 (2.8) |
| HIV status | Reactive | 5 (1.3) | 1 (0.2) |
| | Non-reactive | 187 (47.5) | 14 (3.5) |
| | Unknown | 157 (39.9) | 30 (7.6) |

## Comparison of median survival time

To compare the significance difference in median survival times of various categorical predictors, the log-rank test was used. Accordingly, there was a significant difference in median survival time between categorical predictors such as weight for age, height for age,

## Treatment outcome for children with severe pneumonia

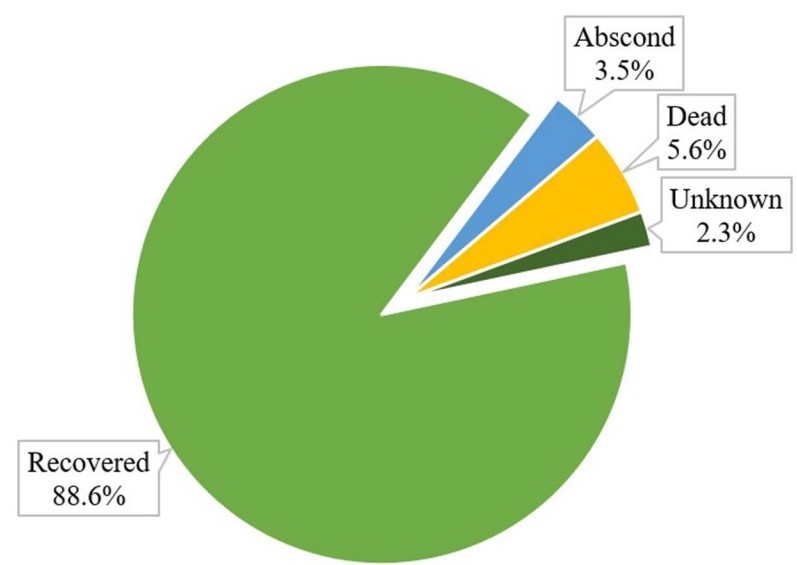

**Fig 2. Treatment outcome for children with severe pneumonia who were admitted to the pediatric ward of JUMC.**

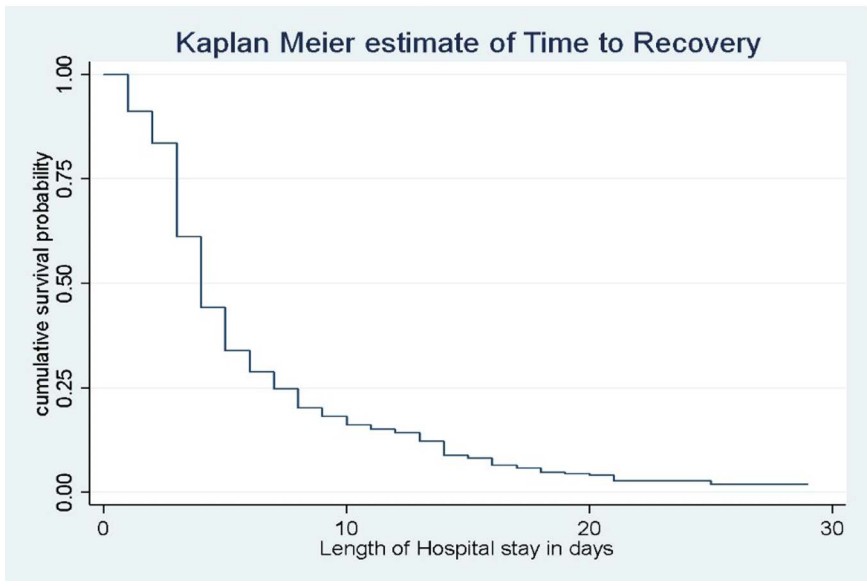

**Fig 3. Kaplan-Meier survival estimate of recovery time among children with severe pneumonia admitted to JUMC from 2018-2022.**

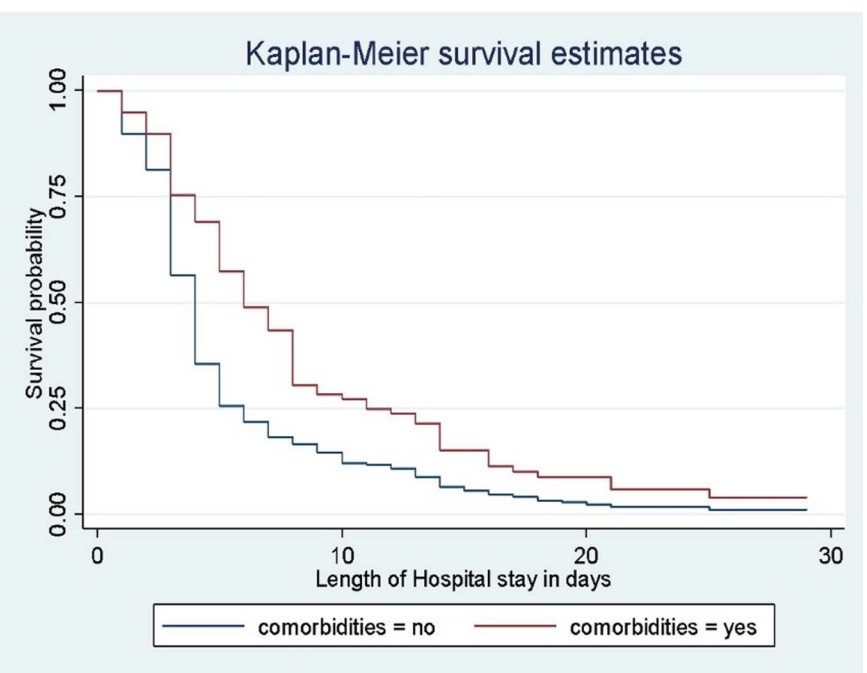

**Fig.4. Kaplan-Meier survival estimate for time to recovery among severe pneumonia children with comorbidity.**

co-morbidity, antibiotics first administered, and antibiotics change. For instance, the median recovery time of children without co-morbidities was 4 days (IQR, 3–6) but for children with co-morbidities, it was 6 days (IQR, 4–11). Children who visited the Hospital within three days of onset of illness had a median recovery time of 4 days (IQR, 2–5) while those who were visited after seven days was 6 days (IQR, 4–13). For children for whom antibiotics were changed, the median recovery time was 5 days (IQR, 3–9) whereas for children for whom no antibiotics were changed, the median recovery time was 4 days (IQR, 3–6) but, there was no significant variation of median recovery time among categorical predictors like HIV status, residence, grunting, and age.

## Proportional hazard assumption test

The Cox proportional hazard assumption was checked statistically (Global test; p-value = 0.37 indicating the Cox proportional hazard assumption was not violated). Finally, Cox-Snell residual test was performed to check the final model adequacy and confirmed that the model is good to fit the data (Fig 5).

## Predictors of time to recovery from severe pneumonia

After running univariate Cox regression, predictors such as sex, age, residence, grunting, antibiotic first administered, antibiotic change, vaccination status, height for age, weight for age, weight for height, co-morbidities, and duration of time before seeking healthcare were selected as a candidate variable for multivariable Cox regression analysis. After performing multivariable Cox regression, adjusted hazard ratios with a 95% confidence interval (CI) and a p-value of less than or equal to 0.05 were used to declare statistical significance. Thus, Variables such as presence of co-morbidities, duration of time before seeking health care, antibiotic first administered and antibiotics change were statistically significant predictors of time to recovery from severe pneumonia in this study.

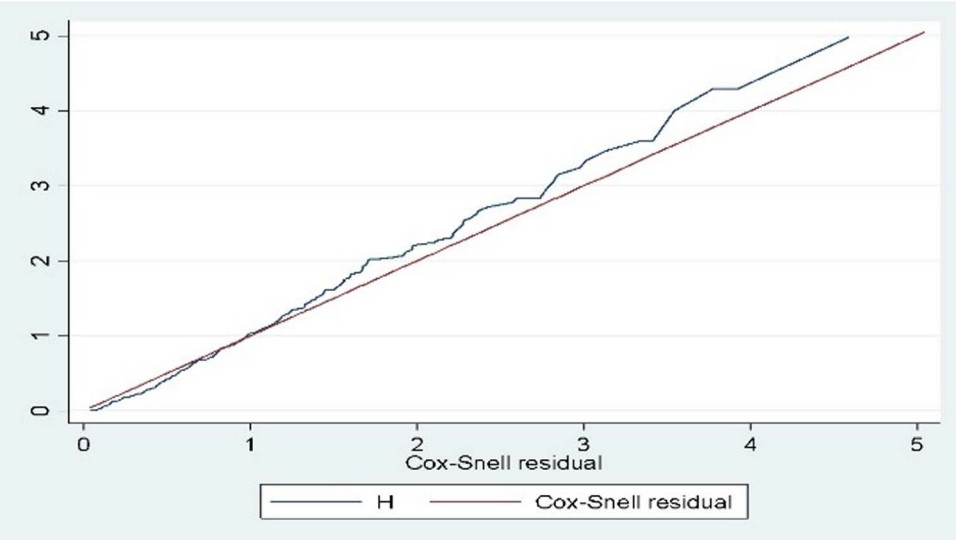

**Fig 5. A cumulative hazard plot of cox Snell residuals for model fitness.**

The recovery time of children with comorbidity is delayed by 30% as compared to children without comorbidity (AHR; 0.70, (95% CI, (0.54–0.92)). Regarding the treatment regimen, the recovery time of children treated with Ceftazidime and Vancomycin as first-line antibiotics during admission was delayed by 71% as compared to children treated with crystalline penicillin (AHR; 0.29, (95% CI, 0.145–0.614)). Children for whom antibiotics were changed were 26% less likely to recover quickly than children for whom antibiotics were not changed (AHR; 0.74, (95% CI, 0.57–0.94)). Children with severe pneumonia who came between 4 and 7 days to the Hospital had 27% slower recovery time as compared to children who came within 3 days (AHR; 0.73(95% CI, (0.561–0.959)) and those who came after 7 days of their illness to the Hospital had 42% slower recovery time as compared to the reference group (AHR; 0.58, (95% CI, 0.43–0.79)) (Table 3).

## Discussion

Children between the ages of 2–59 months with severe pneumonia who were admitted to the pediatric ward of Jimma University Medical Center (JUMC) were assessed for recovery time and its predictors. In the current study, the median recovery time from severe pneumonia was 4 days IQR (3, 7). Different predictors were found to contribute to the recovery time of children admitted with severe pneumonia in this study. Predictors like antibiotic first administered, antibiotic change, duration of time before seeking health care, and comorbidity had significant associations with recovery time.

According to the current study, among children who were admitted to the pediatric ward of Jimma University Medical Center between the ages of 2–59 months, the median recovery time from severe pneumonia was 4 days. This finding is in line with other studies conducted in Debre Markos [29], Hosanna [8] and Gambia [12]. This finding is longer than the British Thoracic Society guideline [30] and the study conducted in Nepal that estimates the recovery time of 2 days [31]. The variation in the results may be explained by the difference in socio-economic status of the research areas, unique circumstances, such as health care setting, and the age of the study participants, as the study involved Nepalese children between the ages of 2–35 months.

The current finding is lower than a study conducted in Tanzania which estimated the median recovery time of 10 days. [32] The discrepancy could be a result of the time difference between study periods. The difference in the age of the study population could also explain the discrepancy, as the Tanzanian study was conducted among children aged between the ages of 0 and 144 months. The analysis approach (model) used, in the Tanzanian study was logistic regression, which does not take the censoring of study participants into account. The study design in Tanzanian study was a prospective cohort study which is better in generating strong evidence could explain the variability.

According to the study conducted in Poland, patients stay for an average of 8.2 to 10.1 days [33]. This variation might be due to high bed occupancy rate in Ethiopia and patients are discharged, as soon as their signs and symptoms subside without sending and waiting for any laboratory confirmation. On the contrary, Poland is high-income country compared to Ethiopia, and their health care system might be well organized and developed [34]. The bed occupancy rate in Poland is probably low and people can pay any charge at the hospital that may create an opportunity for patients to stay longer at the hospital until complete stability from all signs and symptoms of the disease.

**Table 3. Multivariable Cox regression analysis for independent predictors of recovery time among children aged 2-59 months with severe pneumonia admitted to the pediatric ward of Jimma University Medical Center (JUMC) from Jan 1, 2018- Dec 31, 2022.**

| Variable | Category | p-value | CHR, 95%CI | AHR, 95%CI |
|---|---|---|---|---|
| Sex | Male<br>Female | 0.038 | 1<br>0.79(0.645–0.988) | 1<br>0.92(0.727 - 1.155) |
| Residence | Urban<br>Rural | 0.225 | 1<br>0.87(0.711–1.083) | 1<br>0.94 (0.752 - 1.175) |
| Age | 2–11 months<br>12–35 months<br>36–59 months | 0.226<br>0.204 | 1<br>0.86(0.684–1.093)<br>0.80(0.576–1.124) | 1<br>1.00(0.736 - 1.382)<br>1.05(0.678 - 1.636) |
| Grunting | No<br>Yes | 0.122 | 1<br>0.84(0.686–1.045) | 1<br>0.85 (0.680 - 1.071) |
| Antibiotics first administered | Cr. penicillin<br>Ampicillin & Gentamycin<br>Ceftriaxone<br>Ceftazidime & Vancomycin | 0.000<br>0.000<br>0.000 | 1<br>0.50(0.379–0.665)<br>0.55(0.432–0.714)<br>0.23(0.117–0.457) | 1<br>**0.59(0.439- 0.798) \*\***<br>**0.56(0.426 - 0.727) \*\***<br>**0.29(0.143- 0.600) \*\*** |
| Antibiotic change | No<br>Yes | 0.001 | 1<br>0.66(0.532–0.839) | 1<br>**0.74(0.581 - 0.953) \*** |
| Vaccination status | Fully vaccinated<br>Partially vaccinated<br>unvaccinated | 0.734<br>0.001 | 1<br>1.04(0.829–1.303)<br>0.57(0.411–0.799) | 1<br>1.01 (0.728 - 1.402)<br>0.70(0.489 - 1.029) |
| Time prior to seeking health care | Up to 3 days<br>4 - 7days<br>Above 7 days | 0.003<br>0.000 | 1<br>0.67(0.525–0.872)<br>0.51(0.392–0.669) | 1<br>**0.73 (0.561- 0.959) \***<br>**0.58 (0.432 - 0.789) \*\*** |
| Comorbidities | No<br>Yes | 0.000 | 1<br>0.60(0.473-0.774) | 1<br>**0.70 (0.543 - 0.914) \*** |
| Height for age | Normal<br>Stunted | 0.059 | 1<br>0.81(0.651-1.007) | 1<br>0.85(0.671- 1.072) |
| Weight for height | Normal<br>Wasted | 0.055 | 1<br>1.23(.995-1.528) | 1<br>1.15(0.919 - 1.439) |
| Weight for age | Normal<br>Underweight | 0.003 | 1<br>0.71(0.576-0.890) | 1<br>0.89(0.699 - 1.134) |

\*P-Value ≤ 0.05 and

\*\*for p-value < 0.01

Duration of time prior to seeking health care was an independent significant predictor of the recovery time of severe pneumonia. Children who came to the hospital within three days of illness recovered quicker than those children who came after seven days of illness. This finding is supported by a prospective follow-up study conducted in the Gambia [12] and other retrospective studies conducted in Debre Markos Hospital [29], Ayder Comprehensive Specialized Hospital [22], and Central and North Gondar zones [21]. Lately presented cases of pneumonia might develop a complication that needs further investigation to treat and this condition in turn increases the length of time required to recover [35].

The existence of co-morbidity was another significant predictor that was connected with the length of time needed to recover from severe pneumonia in this study. Those children with any type of co-morbidity were found to stay longer in the hospital than their counterparts who were free of co-morbidities. This finding is consistent with other studies conducted at Debre Markos [29], Central and North Gondar zones [21], and Nepal [31]. The reason why children with co-morbidity stay for a prolonged time at the hospital might be due to the fact that co-morbidities significantly increase the severity of pneumonia, including the need for oxygen, and are significantly associated with more complications, that increases the recovery time [32]. Since the major comorbidity in the current study was severe acute malnutrition, as adequate protein and energy are needed for proper immune system functioning; undernourished children have a compromised immune system and are more vulnerable to developing serious infections and complications [36].

The antibiotic first administered was one of the significant clinical and treatment-related predictors in this study. Children with severe pneumonia who were treated with ceftriaxone and ampicillin and gentamicin had a prolonged recovery time. This might be due to ampicillin and gentamicin being administered together for patients presented with severe acute malnutrition in addition to severe pneumonia [37]. Children with severe pneumonia who were treated with Ceftazidime and vancomycin as the first antibiotic at the time of admission stayed longer in the hospital. This might be because Ceftazidime and Vancomycin together administered for children with comorbidity, particularly for severe bacterial infections like endocarditis and invasive methicillin- resistance staphylococcus aureus infections [38, 39].

Antibiotic change is also a significant predictor of recovery time. Compared to children whose medications were not altered, those whose antibiotics were changed required longer hospital stays to recover from severe pneumonia. Antibiotics are altered whenever there is treatment failure on first-line drugs due to drug resistance or selection of inappropriate drug to treat the patients [40]. For this reason, the additional time required for second-line drugs to act could explain the longer recovery time. This finding is supported by other study conducted in Hosanna [8].

## Limitations of the Study

Although this study uses five years of data, important variables like parent's income (Wealth index) and educational status were not included. The study also didn't take any laboratory investigation into account such as baseline complete blood count, blood culture, and erythrocyte sedimentation rate.

## Conclusion

This study assessed the recovery time from severe pneumonia and its predictors among children aged 2–59 months admitted to the pediatric ward of Jimma University Medical Center (JUMC). The finding of this study revealed that the median recovery time was longer than other similar studies. Late presentation to hospital, antibiotic first administered, antibiotic

change and presence of co-morbidities were significant predictors that prolonged recovery time. Healthcare providers should have to give due attention to the identified predictors of recovery time. Measures that help children to recover quickly from their illness should be strengthened.

## Ethical considerations

Prior to data collection, this research protocol has been submitted to the Institutional Review Board (IRB) of Jimma University and meets the ethical and scientific standards outlined in national and international guidelines. Jimma University, Institute of Health (Reference No JUIH/IRB/361/23) approved the consent wavier. Then a letter of permission was written to Jimma University Medical Center managers. Letter to access the patient's chart was obtained from the chief executive officer (CEO) of Jimma University Medical Center and the health management information system focal person verbally consented to use the patient's medical chart for data collection. The study was conducted following the relevant international guidelines, regulations, and principles of Helsinki declaration. Verbal or written consent was not obtained from the study participants due to the fact that the data were collected from patients' medical charts and there was no direct contact between patients and data collectors. Throughout the study, the information obtained from the patient's medical records was kept entirely confidential and only accessed by the research team, and solely for the purpose of the research. For data collection purposes, medical record number were utilized, and no personal identifiers were gathered or utilized in the research report.

## Acknowledgments

The authors would like to acknowledge Jimma University Ethical Review Board for providing the ethical clearance letter and Shenen Gibe General Hospital for their valuable and preliminary support. We also extend our heart full gratitude to all the data collectors and supervisors for their efforts in the data collection and chart review process.

## Author contributions

**Conceptualization:** Getu Girma Bekele.

**Data curation:** Masrie Getnet, Dawit Regassa.

**Formal analysis:** Getu Girma Bekele, Masrie Getnet, Tola Getachew Bekele.

**Investigation:** Dawit Regassa.

**Methodology:** Getu Girma Bekele, Masrie Getnet, Dawit Regassa, Tola Getachew Bekele.

**Software:** Getu Girma Bekele.

**Supervision:** Masrie Getnet, Dawit Regassa, Tola Getachew Bekele.

**Validation:** Getu Girma Bekele, Masrie Getnet, Dawit Regassa, Tola Getachew Bekele.

**Visualization:** Getu Girma Bekele, Tola Getachew Bekele.

**Writing – original draft:** Getu Girma Bekele.

**Writing – review & editing:** Masrie Getnet, Dawit Regassa, Tola Getachew Bekele.

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
