## [Decision Letter · Decision Letter 0]

14 Aug 2024

PONE-D-24-04566Time-to-Recovery from Severe Pneumonia and Its Predictors Among Children 2-59 Months of Age Admitted to Pediatric Ward of Jimma University Medical Center, Southwest Ethiopia, 2023: A Retrospective Cohort Study.PLOS ONE

Dear Dr. Girma Bekele,

Thank you for submitting your manuscript to PLOS ONE. After careful consideration, we feel that it has merit but does not fully meet PLOS ONE’s publication criteria as it currently stands. Therefore, we invite you to submit a revised version of the manuscript that addresses the points raised during the review process.

We look forward to receiving your revised manuscript.

Kind regards,

Tamirat Getachew

Academic Editor

PLOS ONE

3. We note that your Data Availability Statement is currently as follows: [All relevant data are within the manuscript and its Supporting Information files]

Reviewers' comments:

Reviewer's Responses to Questions

**Comments to the Author**

1. Is the manuscript technically sound, and do the data support the conclusions?

Reviewer #1: Yes

Reviewer #2: Yes

2. Has the statistical analysis been performed appropriately and rigorously? 

Reviewer #1: Yes

Reviewer #2: Yes

3. Have the authors made all data underlying the findings in their manuscript fully available?

Reviewer #1: Yes

Reviewer #2: Yes

4. Is the manuscript presented in an intelligible fashion and written in standard English?

Reviewer #1: Yes

Reviewer #2: Yes

5. Review Comments to the Author

Reviewer #1: Review Report

Abstract: Inadequately clear

Background: Needs further enrichment for relevance and contextual factors.

Methods: Should be presented in explicit and comprehensive manner including the ethical consideration.

Results and discussion: Lacks brief and comprehensive presentation with appropriate references and justification.

Regards,

Reviewer #2: 1. The manuscript is technically sound, and the data do support the conclusions.

2. The statistical analysis has been performed appropriately and rigorously.

3. The authors have made all data underlying the findings in their manuscript fully available.

4. The manuscript is presented in an intelligible fashion and written in standard English.

6. PLOS authors have the option to publish the peer review history of their article (what does this mean? ). If published, this will include your full peer review and any attached files.

**Do you want your identity to be public for this peer review?** For information about this choice, including consent withdrawal, please see our Privacy Policy .

Reviewer #1: No

Reviewer #2: **Yes: ** Kassa Demissie Abdi (PhD)

---

## [Author Response · Author response to Decision Letter 1]

29 Nov 2024

PLOS ONE| Editorial Office - Collaborative Peer Review Team Comments to Author:

Independent Review Report, Reviewer 1

Abstract: inadequately clear

Response: We appreciate the comments and suggestions; the recommendation you raised were briefly written but we deleted them for the sake of word limit and now we have made corrections in the specified section as per the comments.

Background: needs further enrichment for relevance and contextual factors

Response: we have made corrections in the specified section as per the comments.

Methods: should be presented in explicit and comprehensive manner including ethical considerations.

Response: Thank you for your constructive comments and we made appropriate corrections in the specified section as per the comments

Operational definition: Good if all are referenced with WHO standards or any other = optional

Response: Thank you. We made corrections in the section

Survival time: What is its difference from that of time-to-recovery?

Response: Thank you for the comments, survival time is the general term that indicates the time it takes for the certain event(out come of interest) to occur and our event in this particular study was recovery. Indeed there is no such difference between them and we have deleted survival time

Do you have graph?

Response: Thank you for the comments, yes we do have graphs like Kaplan Meier survival curve and Cox-snell.

Why? What is unique about residence?

Response: Thank you for the comments, in this particular study event(recovery) was higher than censoring but we made a mistake about residence during typing (simply typing error) and so we made corrections in the section.

Results and discussion: lacks brief and comprehensive presentation with appropriate references and justification

Response: Thank you. We made corrections in the section.

Table 2 Revisit your percentages

Response: Thank you. We made corrections in the section.

Vaccination status: Repeated

Response: Thank you. We mistakenly repeated and we deleted repetition.

After performing multivariable Cox regression, adjusted hazard ratios with a 95% confidence interval (CI) and a p-value of less than or equal to 0.05. Thus, variables like the presence of comorbidities, duration of time before seeking health care, antibiotic first administered and antibiotics change were statistically significant predictors of time to recovery from severe pneumonia in this study. Seems complete?

Response: Thank you. We made corrections in the section as After performing multivariable Cox regression, adjusted hazard ratios with a 95% confidence interval (CI) and a p-value of less than or equal to 0.05 were used to declare statistical significance. Thus, Variables such as presence of co-morbidities, duration of time before seeking health care, antibiotic first administered and antibiotics change were statistically significant predictors of time to recovery from severe pneumonia in this study.

Table 3: Age did not contribute. Why?

Response: Thank you. After controlling the potential confounding factors, there is no siginficant difference in the age between 2-11 months and 12-35 months, indicating that the crude effect was likely not due to the factors being studied, but rather the other confounding factors.

In discussion part:

Have you read any single article that tells you this information?

Response: Thank you for the comments and we cited the references as per the comments in our manuscript.

Rewrite. It is a long sentence to capture your idea. So, split into two and rewrite it again

Rewrite in a clear way

Response: Thank you. We made corrections in the section.

---

## [Decision Letter · Decision Letter 1]

17 Dec 2024

Time-to-Recovery from Severe Pneumonia and Its Predictors Among Children 2-59 Months of Age Admitted to Pediatric Ward of Jimma University Medical Center, Southwest Ethiopia, 2023: A Retrospective Cohort Study.

PONE-D-24-04566R1

Dear Getu,

We’re pleased to inform you that your manuscript has been judged scientifically suitable for publication and will be formally accepted for publication once it meets all outstanding technical requirements.

Kind regards,

Tamirat Getachew

Academic Editor

PLOS ONE

Additional Editor Comments (optional):

Reviewers' comments:

Reviewer's Responses to Questions

**Comments to the Author**

1. If the authors have adequately addressed your comments raised in a previous round of review and you feel that this manuscript is now acceptable for publication, you may indicate that here to bypass the “Comments to the Author” section, enter your conflict of interest statement in the “Confidential to Editor” section, and submit your "Accept" recommendation.

Reviewer #2: All comments have been addressed

2. Is the manuscript technically sound, and do the data support the conclusions?

Reviewer #2: Yes

3. Has the statistical analysis been performed appropriately and rigorously? 

Reviewer #2: Yes

4. Have the authors made all data underlying the findings in their manuscript fully available?

Reviewer #2: Yes

5. Is the manuscript presented in an intelligible fashion and written in standard English?

Reviewer #2: Yes

6. Review Comments to the Author

Reviewer #2: 1. All comments have been addressed.

2. The manuscript is technically sound, and the data do support the conclusions.

3. The statistical analysis has been performed appropriately and rigorously.

4. The authors have made all data underlying the findings in their manuscript fully available.

5. The manuscript is presented in an intelligible fashion and written in standard English.

7. PLOS authors have the option to publish the peer review history of their article (what does this mean? ). If published, this will include your full peer review and any attached files.

**Do you want your identity to be public for this peer review?** For information about this choice, including consent withdrawal, please see our Privacy Policy .

Reviewer #2: **Yes: ** Kassa Demissie Abdi (PhD)

---

## [Editor Report · Acceptance letter]

PONE-D-24-04566R1

PLOS ONE

Dear Dr. Girma Bekele,

I'm pleased to inform you that your manuscript has been deemed suitable for publication in PLOS ONE. Congratulations! Your manuscript is now being handed over to our production team.

Kind regards,

on behalf of

Dr. Tamirat Getachew

Academic Editor

PLOS ONE